# The Expressive Power of Neural Networks: A View from the Width

**Zhou Lu**[1,3]
1400010739@pku.edu.cn

**Hongming Pu**[1]
1400010621@pku.edu.cn

**Feicheng Wang**[1,3]
1400010604@pku.edu.cn

**Zhiqiang Hu**[2]
huzq@pku.edu.cn

**Liwei Wang**[2,3]
wanglw@cis.pku.edu.cn

1, Department of Mathematics, Peking University
2, Key Laboratory of Machine Perception, MOE, School of EECS, Peking University
3, Center for Data Science, Peking University, Beijing Institute of Big Data Research

## Abstract

The expressive power of neural networks is important for understanding deep learning. Most existing works consider this problem from the view of the depth of a network. In this paper, we study how width affects the expressiveness of neural networks. Classical results state that *depth-bounded* (e.g. depth-2) networks with suitable activation functions are universal approximators. We show a universal approximation theorem for *width-bounded* ReLU networks: width-$(n + 4)$ ReLU networks, where $n$ is the input dimension, are universal approximators. Moreover, except for a measure zero set, all functions cannot be approximated by width-$n$ ReLU networks, which exhibits a phase transition. Several recent works demonstrate the benefits of depth by proving the depth-efficiency of neural networks. That is, there are classes of deep networks which cannot be realized by any shallow network whose size is no more than an *exponential* bound. Here we pose the dual question on the width-efficiency of ReLU networks: Are there wide networks that cannot be realized by narrow networks whose size is not substantially larger? We show that there exist classes of wide networks which cannot be realized by any narrow network whose depth is no more than a *polynomial* bound. On the other hand, we demonstrate by extensive experiments that narrow networks whose size exceed the polynomial bound by a constant factor can approximate wide and shallow network with high accuracy. Our results provide more comprehensive evidence that depth may be more effective than width for the expressiveness of ReLU networks.

## 1  Introduction

Deep neural networks have achieved state-of-the-art performance in a wide range of tasks such as speech recognition, computer vision, natural language processing, and so on. Despite their promising results in applications, our theoretical understanding of neural networks remains limited. The expressive power of neural networks, being one of the vital properties, is crucial on the way towards a more thorough comprehension.

The expressive power describes neural networks' ability to approximate functions. This line of research dates back at least to 1980's. The celebrated universal approximation theorem states that depth-2 networks with suitable activation function can approximate any continuous function on a compact domain to any desired accuracy [3] [1] [9] [6]. However, the size of such a neural network

can be exponential in the input dimension, which means that the depth-2 network has a very large width.

From a learning perspective, having universal approximation is just the first step. One must also consider the efficiency, i.e., the size of the neural network to achieve approximation. Having a small size requires an understanding of the roles of depth and width for the expressive power. Recently, there are a series of works trying to characterize how depth affects the expressiveness of a neural network . [5] showed the existence of a 3-layer network, which cannot be realized by any 2-layer to more than a constant accuracy if the size is subexponential in the dimension. [2] proved the existence of classes of deep convolutional ReLU networks that cannot be realized by shallow ones if its size is no more than an exponential bound. For any integer $k$, [15] explicitly constructed networks with $O(k^3)$ layers and constant width which cannot be realized by any network with $O(k)$ layers whose size is smaller than $2^k$. This type of results are referred to as depth efficiency of neural networks on the expressive power: a reduction in depth results in exponential sacrifice in width. However, it is worth noting that these are existence results. In fact, as pointed out in [2], proving existence is inevitable; There is always a positive measure of network parameters such that deep nets can't be realized by shallow ones without substantially larger size. Thus we should explore more in addition to proving existence.

Different to most of the previous works which investigate the expressive power in terms of the depth of neural networks, in this paper we study the problem from the view of *width*. We argue that an integration of both views will provide a better understanding of the expressive power of neural networks.

Firstly, we prove a universal approximation theorem for width-bounded ReLU networks. Let $n$ denotes the input dimension, we show that width-$(n + 4)$ ReLU networks can approximate any Lebesgue integrable function on $n$-dimensional space with respect to $L^1$ distance. On the other hand, except for a zero measure set, all Lebesgue integrable functions cannot be approximated by width-$n$ ReLU networks, which demonstrate a phase transition. Our result is a dual version of the classical universal approximation theorem for depth-bounded networks.

Next, we explore quantitatively the role of width for the expressive power of neural networks. Similar to the depth efficiency, we raise the following question on the width efficiency:

*Are there wide ReLU networks that cannot be realized by any narrow network whose size is not substantially increased?*

We argue that investigation of the above question is important for an understanding of the roles of depth and width for the expressive power of neural networks. Indeed, if the answer to this question is *yes*, and the size of the narrow networks must be *exponentially* larger, then it is appropriate to say that width has an equal importance as depth for neural networks.

In this paper, we prove that there exists a family of ReLU networks that cannot be approximated by narrower networks whose depth increase is no more than *polynomial*. This polynomial lower bound for width is significantly smaller than the exponential lower bound for depth. However, it does not rule out the possibility of the existence of an exponential lower bound for width efficiency. On the other hand, insights from the previous analysis suggest us to study if there is a polynomial upper bound, i.e., a polynomial increase in depth and size suffices for narrow networks to approximate wide and shallow networks. Theoretically proving a polynomial upper bound seems very difficult, and we formally pose it as an open problem. Nevertheless, we conduct extensive experiments and the results demonstrate that when the depth of the narrow network exceeds the polynomial lower bound by just a constant factor, it can approximate wide shallow networks to a high accuracy. Together, these results provide more comprehensive evidence that depth is more effective for the expressive power of ReLU networks.

Our contributions are summarized as follows:

- We prove a Universal Approximation Theorem for Width-Bounded ReLU Networks. We show that any Lebesgue-integrable function $f$ from $\mathbb{R}^n$ to $\mathbb{R}$ can be approximated by a fully-connected width-$(n + 4)$ ReLU network to arbitrary accuracy with respect to $L^1$ distance. In addition, except for a negligible set, all functions $f$ from $\mathbb{R}^n$ to $\mathbb{R}$ cannot be approximated by any ReLU network whose width is no more than $n$.

- We show a width efficiency polynomial lower bound. For integer $k$, there exist a class of width-$O(k^2)$ and depth-2 ReLU networks that cannot be approximated by any width-$O(k^{1.5})$ and depth-$k$ networks. On the other hand, experimental results demonstrate that networks with size slightly larger than the lower bound achieves high approximation accuracy.

## 1.1 Related Work

Research analyzing the expressive power of neural networks date back to decades ago. As one of the most classic work, Cybenko[3] proved that a fully-connected sigmoid neural network with one single hidden layer can universally approximate any continuous univariate function on a bounded domain with arbitrarily small error. Barron[1], Hornik et al.[9] ,Funahashi[6] achieved similar results. They also generalize the sigmoid function to a large class of activation functions, showing that universal approximation is essentially implied by the network structure. Delalleau et al.[4] showed that there exists a family of functions which can be represented much more efficiently with deep networks than with shallow ones as well.

Due to the development and success of deep neural networks recently, there have been much more works discussing the expressive power of neural networks theoretically. Depth efficiency is among the most typical results. Eldan et.al [5] showed the existence of a 3-layer network, which cannot be realized by any 2-layer to more than a constant accuracy if the size is subexponential in the dimension. Cohen et.al [2] proved the existence of classes of deep convolutional ReLU networks that cannot be realized by shallow ones if its size is no more than an exponential bound. For any integer $k$, Telgarsky [15] explicitly constructed networks with $O(k^3)$ layers and constant width which cannot be realized by any network with $O(k)$ layers whose size is smaller than $2^k$.

Other works turn to show deep networks' ability to approximate a wide range of functions. For example, Liang et al.[12] showed that in order to approximate a function which is $\Theta(\log \frac{1}{\epsilon})$-order derivable with $\epsilon$ error universally, a deep network with $O(\log \frac{1}{\epsilon})$ layers and $O(\text{poly} \log \frac{1}{\epsilon})$ weights can do but $\Omega(\text{poly} \frac{1}{\epsilon})$ weights will be required if there is only $o(\log \frac{1}{\epsilon})$ layers. Yarotsky [16] showed that $C^n$-functions on $\mathbb{R}^d$ with a bounded domain can be approximated with $\epsilon$ error universally by a ReLU network with $O(\log \frac{1}{\epsilon})$ layers and $O((\frac{1}{\epsilon})^{\frac{d}{n}} \log \frac{1}{\epsilon})$ weights. In addition, for results based on classic theories, Harvey et al.[7] provided a nearly-tight bound for VC-dimension of neural networks, that the VC-dimension for a network with $W$ weights and $L$ layers will have a $O(WL \log W)$ but $\Omega(WL \log \frac{W}{L})$ VC-dimension. Also, there are several works arguing for width's importance from other aspects, for example, Nguyen et al.[11] shows if a deep architecture is at the same time sufficiently wide at one hidden layer then it has a well-behaved loss surface in the sense that almost every critical point with full rank weight matrices is a global minimum from the view of optimization.

The remainder of the paper is organized as follows. In section 2 we introduce some background knowledge needed in this article. In section 3 we present our main result – the Width-Bounded Universal Approximation Theorem; besides, we show two comparing results related to the theorem. Then in section 4 we turn to explore quantitatively the role of width for the expressive power of neural networks. Finally, section 5 concludes. All proofs can be found in the Appendix and we give proof sketch in main text as well.

## 2 Preliminaries

We begin by presenting basic definitions that will be used throughout the paper. A neural network is a directed computation graph, where the nodes are computation units and the edges describe the connection pattern among the nodes. Each node receives as input a weighted sum of activations flowed through the edges, applies some kind of activation function, and releases the output via the edges to other nodes. Neural networks are often organized in layers, so that nodes only receive signals from the previous layer and only release signals to the next layer. A fully-connected neural network is a layered neural network where there exists a connection between every two nodes in adjacent layers. In this paper, we will study the fully-connected ReLU network, which is a fully-connected neural

network with Rectifier Linear Unit (ReLU) activation functions. The ReLU function $\mathrm{ReLU}\colon \mathbb{R} \to \mathbb{R}$ can be formally defined as

$$\mathrm{ReLU}(x) = \max\{x, 0\} \tag{1}$$

The architecture of neural networks often specified by the width and the depth of the networks. The depth $h$ of a network is defined as its number of layers (including output layer but excluding input layer); while the width $d_m$ of a network is defined to be the maximal number of nodes in a layer. The number of input nodes, i.e. the input dimension, is denoted as $n$.

In this paper we study the expressive power of neural networks. The expressive power describes neural networks' ability to approximate functions. We focus on Lebesgue-integrable functions. A Lebesgue-integrable function $f\colon \mathbb{R}^n \to \mathbb{R}$ is a Lebesgue-measurable function satisfying

$$\int_{\mathbb{R}^n} |f(x)|\mathrm{d}x < \infty \tag{2}$$

which contains continuous functions, including functions such as the sgn function. Because we deal with Lebesgue-integrable functions, we adopt $L^1$ distance as a measure of approximation error, different from $L^\infty$ distance used by some previous works which consider continuous functions.

## 3 Width-bounded ReLU Networks as Universal Approximator

In this section we consider universal approximation with width-bounded ReLU networks. The following theorem is the main result of this section.

**Theorem 1** (Universal Approximation Theorem for Width-Bounded ReLU Networks)**.** *For any Lebesgue-integrable function $f\colon \mathbb{R}^n \to \mathbb{R}$ and any $\epsilon > 0$, there exists a fully-connected ReLU network $\mathscr{A}$ with width $d_m \le n + 4$, such that the function $F_{\mathscr{A}}$ represented by this network satisfies*

$$\int_{\mathbb{R}^n} |f(x) - F_{\mathscr{A}}(x)|\mathrm{d}x < \epsilon. \tag{3}$$

The proof of this theorem is lengthy and is deferred to the supplementary material. Here we provide an informal description of the high level idea.

For any Lebesgue integrable function and any predefined approximation accuracy, we explicitly construct a width-$(n + 4)$ ReLU network so that it can approximate the function to the given accuracy. The network is a concatenation of a series of blocks. Each block satisfies the following properties:

1) It is a depth-$(4n + 1)$ width-$(n + 4)$ ReLU network.

2) It can approximate any Lebesgue integrable function which is uniformly zero outside a cube with length $\delta$ to a high accuracy;

3) It can store the output of the previous block, i.e., the approximation of other Lebesgue integrable functions on different cubes;

4) It can sum up its current approximation and the memory of the previous approximations.

It is not difficult to see that the construction of the whole network is completed once we build the blocks. We illustrate such a block in Figure 1 . In this block, each layer has $n + 4$ neurons. Each rectangle in Figure 1 represents a neuron, and the symbols in the rectangle describes the output of that neuron as a function of the block. Among the $n + 4$ neurons, $n$ neurons simply transfer the input coordinates. For the other 4 neurons, 2 neurons store the approximation fulfilled by previous blocks. The other 2 neurons help to do the approximation on the current cube. The topology of the block is rather simple. It is very sparse, each neuron connects to at most 2 neurons in the next layer.

The proof is just to verify the construction illustrated in Figure 1 is correct. Because of the space limit, we defer all the details to the supplementary materials.

Theorem 1 can be regarded as a dual version of the classical universal approximation theorem, which proves that depth-bounded networks are universal approximator. If we ignore the size of the network,

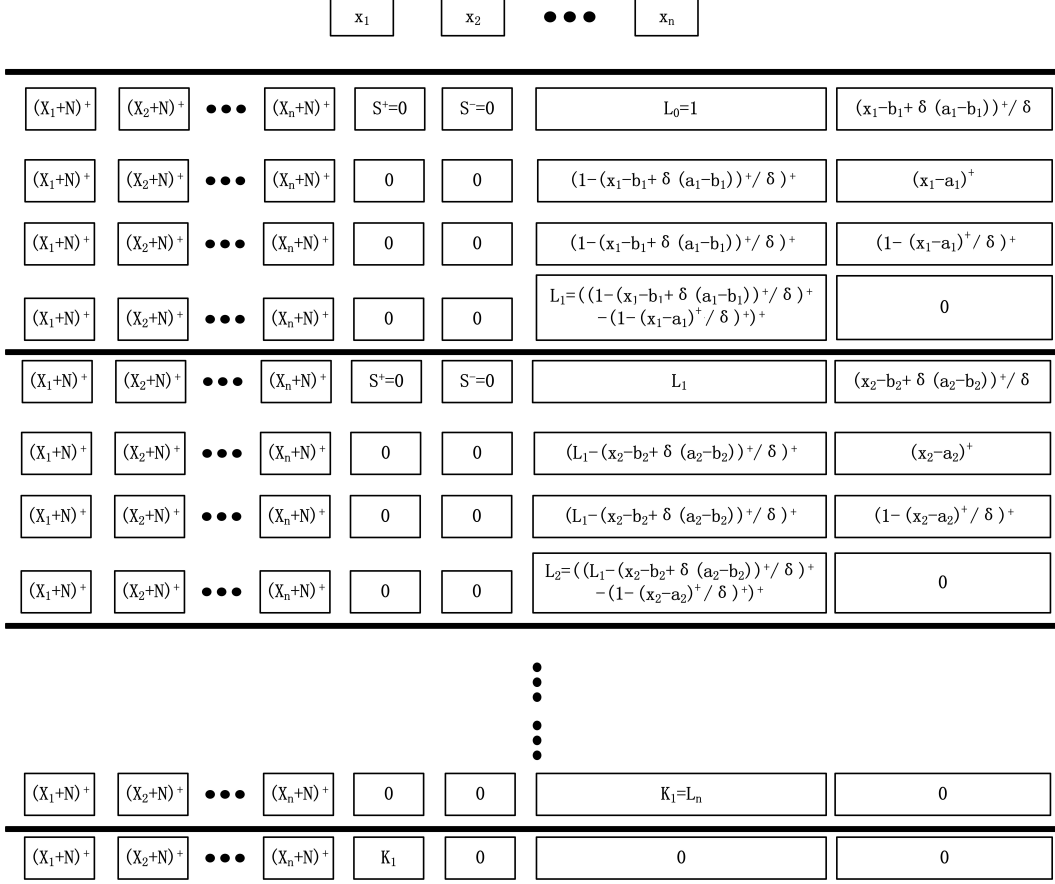

Figure 1: One block to simulate the indicator function on $[a_1, b_1] \times [a_2, b_2] \times \cdots \times [a_n, b_n]$. For $k$ from 1 to $n$, we "chop" two sides in the $kth$ dimension, and for every $k$ the "chopping" process is completed within a 4-layer sub-network as we show in Figure 1. It is stored in the (n+3)th node as $L_n$ in the last layer of $\mathscr{A}$. We then use a single layer to record it in the (n+1)th or the (n+2)th node, and reset the last two nodes to zero. Now the network is ready to simulate another (n+1)-dimensional cube.

both depth and width themselves are efficient for universal approximation. At the technical level however, there are a few differences between the two universal approximation theorems. The classical depth-bounded theorem considers continuous function on a compact domain and use $L^\infty$ distance; Our width-bounded theorem instead deals with Lebesgue-integrable functions on the whole Euclidean space and therefore use $L^1$ distance.

Theorem 1 implies that there is a phase transition for the expressive power of ReLU networks as the width of the network varies across $n$, the input dimension. It is not difficult to see that if the width is much smaller than $n$, then the expressive power of the network must be very weak. Formally, we have the following two results.

**Theorem 2.** *For any Lebesgue-integrable function $f : \mathbb{R}^n \to \mathbb{R}$ satisfying that $\{x : f(x) \neq 0\}$ is a positive measure set in Lebesgue measure, and any function $F_\mathscr{A}$ represented by a fully-connected ReLU network $\mathscr{A}$ with width $d_m \leq n$, the following equation holds:*

$$\int_{\mathbb{R}^n} |f(x) - F_\mathscr{A}(x)| \mathrm{d}x = +\infty \ or \int_{\mathbb{R}^n} |f(x)| \mathrm{d}x. \tag{4}$$

Theorem 2 says that even the width equals $n$, the approximation ability of the ReLU network is still weak, at least on the Euclidean space $\mathbb{R}^n$. If we restrict the function on a bounded set, we can still prove the following theorem.

**Theorem 3.** *For any continuous function $f : [-1, 1]^n \to \mathbb{R}$ which is not constant along any direction, there exists a universal $\epsilon^* > 0$ such that for any function $F_A$ represented by a fully-connected ReLU network with width $d_m \leq n - 1$, the $L^1$ distance between $f$ and $F_A$ is at least $\epsilon^*$:*

$$\int_{[-1,1]^n} |f(x) - F_A(x)| \mathrm{d}x \geq \epsilon^*. \tag{5}$$

Then Theorem 3 is a direct comparison with Theorem 1 since in Theorem 1 the $L^1$ distance can be arbitrarily small.

The main idea of the two theorems is grabbing the disadvantage brought by the insufficiency of dimension. If the corresponding first layer values of two different input points are the same, the output will be the same as well. When the ReLU network's width is not larger than the input layer's width, we can find a ray for "most" points such that the ray passes the point and the corresponding first layer values on the ray are the same. It is like a dimension reduction caused by insufficiency of width. Utilizing this weakness of thin network, we can finally prove the two theorems.

# 4   Width Efficiency vs. Depth Efficiency

Going deeper and deeper has been a trend in recent years, starting from the 8-layer AlexNet[10], the 19-layer VGG[13], the 22-layer GoogLeNet[14], and finally to the 152-layer and 1001-layer ResNets[8]. The superiority of a larger depth has been extensively shown in the applications of many areas. For example, ResNet has largely advanced the state-of-the-art performance in computer vision related fields, which is claimed solely due to the extremely deep representations. Despite of the great practical success, theories of the role of depth are still limited.

Theoretical understanding of the strength of depth starts from analyzing the depth efficiency, by proving the existence of deep neural networks that cannot be realized by any shallow network whose size is exponentially larger. However, we argue that even for a comprehensive understanding of the depth itself, one needs to study the dual problem of width efficiency: Because, if we switch the role of depth and width in the depth efficiency theorems and the resulting statements remain true, then width would have the same power as depth for the expressiveness, at least in theory. It is worth noting that a priori, depth efficiency theorems do not imply anything about the validity of width efficiency.

In this section, we study the width efficiency of ReLU networks quantitatively.

**Theorem 4.** *Let $n$ be the input dimension. For any integer $k \geq n + 4$, there exists $F_{\mathscr{A}} : \mathbb{R}^n \to \mathbb{R}$ represented by a ReLU neural network $\mathscr{A}$ with width $d_m = 2k^2$ and depth $h = 3$, such that for any constant $b > 0$, there exists $\epsilon > 0$ and for any function $F_{\mathscr{B}} : \mathbb{R}^n \to \mathbb{R}$ represented by ReLU neural network $\mathscr{B}$ whose parameters are bounded in $[-b, b]$ with width $d_m \leq k^{3/2}$ and depth $h \leq k + 2$, the following inequality holds:*

$$\int_{\mathbb{R}^n} |F_{\mathscr{A}} - F_{\mathscr{B}}| \mathrm{d}x \geq \epsilon. \tag{6}$$

Theorem 4 states that there are networks such that reducing width requires increasing in the size to compensate, which is similar to that of depth qualitatively. However, at the quantitative level, this theorem is very different to the depth efficiency theorems in [15] [5][2]. Depth efficiency enjoys exponential lower bound, while for width Theorem 4 is a polynomial lower bound. Of course if a corresponding polynomial upper bound can be proven, we can say depth plays a more important role in efficiency, but such a polynomial lower bound still means that depth is not strictly stronger than width in efficiency, sometimes it costs depth super-linear more nodes than width.

This raises a natural question: Can we improve the polynomial lower bound? There are at least two possibilities.

1) Width efficiency has *exponential lower bound*. To be concrete, there are wide networks that cannot be approximated by any narrow networks whose size is no more than an exponential bound.

2) Width efficiency has *polynomial upper bound*. Every wide network can be approximated by a narrow network whose size increase is no more than a polynomial.

Exponential lower bound and polynomial upper bound have completely different implications. If exponential lower bound is true, then width and depth have the same strength for the expressiveness,

at least in theory. If the polynomial upper bound is true, then depth plays a significantly stronger role for the expressive power of ReLU networks.

Currently, neither the exponential lower bound nor the polynomial upper bound seems within the reach. We pose it as a formal open problem.

### 4.1 Experiments

We further conduct extensive experiments to provide some insights about the upper bound of such an approximation. To this end, we study a series of network architectures with varied width. For each network architecture, we randomly sample the parameters, which, together with the architecture, represent the function that we would like narrower networks to approximate. The approximation error is empirically calculated as the mean square error between the target function and the approximator function evaluated on a series of uniformly placed inputs. For simplicity and clearity, we refer to the network architectures that will represent the target functions when assigned parameters as target networks, and the corresponding network architectures for approximator functions as approximator networks.

To be detailed, the target networks are fully-connected ReLU networks of input dimension $n$, output dimension 1, width $2k^2$ and depth 3, for $n = 1, 2$ and $k = 3, 4, 5$. For each of these networks, we sample weight parameters according to standard normal distribution, and bias parameters according to uniform distribution over $[-1, 1)$. The network and the sampled parameters will collectively represent a target function that we use a narrow approximator network of width $3k^{3/2}$ and depth $k + 2$ to approximate, with a corresponding $k$. The architectures are designed in accordance to Theorem 4 – we aim to investigate whether such a lower bound is actually an upper bound. In order to empirically calculate the approximation error, 20000 uniformly placed inputs from $[-1, 1)^n$ for $n = 1$ and 40000 such inputs for $n = 2$ are evaluated by the target function and the approximator function respectively, and the mean square error is reported. For each target network, we repeat the parameter-sampling process 50 times and report the mean square error in the worst and average case.

We adopt the standard supervised learning approach to search in the parameter space of the approximator network to find the best approximator function. Specifically, half of all the test inputs from $[-1, 1)^n$ and the corresponding values evaluated by target function constitute the training set. The training set is used to train approximator network with a mini-batch AdaDelta optimizer and learning rate 1.0. The parameters of approximator network are randomly initialized according to [8]. The training process proceeds 100 epoches for $n = 1$ and 200 epoches for $n = 2$; the best approximator function is recorded.

Table 1 lists the results. Figure 2 illustrates the comparison of an example target function and the corresponding approximator function for $n = 1$ and $k = 5$. Note that the target function values vary with a scale $\sim 10$ in the given domain, so the (absolute) mean square error is indeed a rational measure of the approximation error. It is shown that the approximation error is indeed very small, for the target networks and approximator networks we study. From Figure 2 we can see that the approximation function is so close to the target function that we have to enlarge a local region to better display the difference. Since the architectures of both the target networks and approximator networks are determined according to Theorem 4, where the depth of approximator networks are in a polynomial scale with respect to that of target networks, the empirical results show an indication that a polynomial larger depth may be sufficient for a narrow network to approximate a wide network.

## 5 Conclusion

In this paper, we analyze the expressive power of neural networks with a view from the *width*, distinguished from many previous works which focus on the view from the *depth*. We establish the Universal Approximation Theorem for Width-Bounded ReLU Networks, in contrast with the well-known Universal Approximation Theorem, which studies depth-bounded networks. Our result demonstrate a phase transition with respect to expressive power when the width of a ReLU network of given input dimension varies.

We also explore the role of width for the expressive power of neural networks: we prove that a wide network cannot be approximated by a narrow network unless with polynomial more nodes, which gives a lower bound of the number of nodes for approximation. We pose open problems on whether

Table 1: Empirical study results. $n$ denotes the input dimension, $k$ is defined in Theorem 4; the width/depth for both target network and approximator network are determined in accordance to Theorem 4. We report mean square error in the worst and average case over 50 runs of randomly sampled parameters for target network.

| $n$ | $k$ | target network | | approximator network | | worst case error | average case error |
|---|---|---|---|---|---|---|---|
| | | width | depth | width | depth | | |
| 1 | 3 | 18 | 3 | 16 | 5 | 0.002248 | 0.000345 |
| 1 | 4 | 36 | 3 | 24 | 6 | 0.003263 | 0.000892 |
| 1 | 5 | 50 | 3 | 34 | 7 | 0.005643 | 0.001296 |
| 2 | 3 | 18 | 3 | 16 | 5 | 0.008729 | 0.001990 |
| 2 | 4 | 36 | 3 | 24 | 6 | 0.018852 | 0.006251 |
| 2 | 5 | 50 | 3 | 34 | 7 | 0.030114 | 0.007984 |

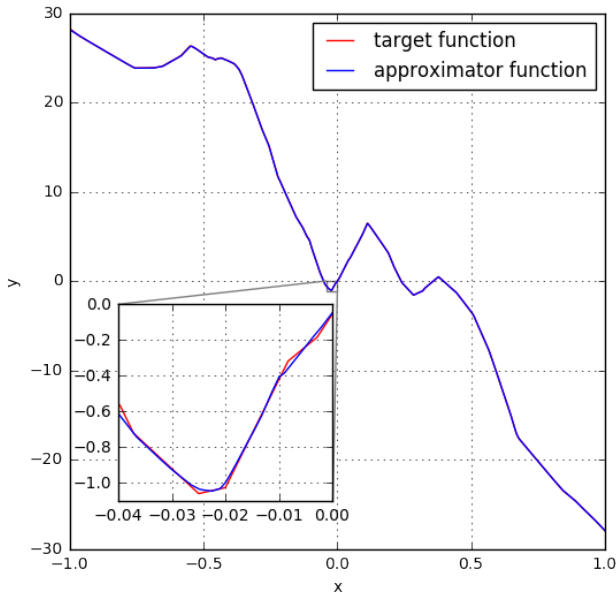

Figure 2: Comparison of an example target function and the corresponding approximator function for $n = 1$ and $k = 5$. A local region is enlarged to better display the difference.

exponential lower bound or polynomial upper bound hold for the width efficiency, which we think is crucial on the way to a more thorough understanding of expressive power of neural networks. Experimental results support the polynomial upper bound and agree with our intuition and insights from the analysis.

The width and the depth are two key components in the design of a neural network architecture. Width and depth are both important and should be carefully tuned together for the best performance of neural networks, since the depth may determine the abstraction level but the width may influence the loss of information in the forwarding pass. A comprehensive understanding of the expressive power of neural networks requires looking from both views.

## Acknowledgments

This work was partially supported by National Basic Research Program of China (973 Program) (grant no. 2015CB352502), NSFC (61573026) and Center for Data Science, Beijing Institute of Big Data Research in Peking University. We would like to thank the anonymous reviewers for their valuable comments on our paper.

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
