[Supplementary Material]

# Appendix

## A    Proof of Theorem 1

*Proof.* We prove this theorem by constructing a network architecture which can approximate any Lesbegue-integrable function w.r.t $L^1$ distance. We will firstly illustrate that $f$ can be approximated by finite weighted sum of indicator functions on n-dimensional cubes. Then we will show how a ReLU network approximate an indicator function on an n-dimensional cube. Finally we will show that ReLU network can "store" the quantities and sum them up.

Assume $x = (x_1, \ldots, x_n)$ is the input. Since $f$ is L-integrable, for any $\epsilon > 0$, there exists $N > 0$ which satisfies

$$\int_{\cup_{i=1}^n |x_i| \geq N} |f| dx < \frac{\epsilon}{2}$$

For simplication, the following symbols are introduced.

$$E \triangleq [-N, N]^n$$

$$f_1(x) \triangleq \begin{cases} max\{f, 0\} & x \in E \\ 0 & x \notin E \end{cases}$$

$$f_2(x) \triangleq \begin{cases} max\{-f, 0\} & x \in E \\ 0 & x \notin E \end{cases}$$

$$C \triangleq \int_{R^n} |f| d\vec{x}$$

$$V_E^1 \triangleq \{(x, y) | x \in E, 0 < y < f_1(x)) \}$$
$$V_E^2 \triangleq \{(x, y) | x \in E, 0 < y < f_2(x)) \}$$

Then we have

$$\int_{R^n} |f - (f_1 - f_2)| dx < \frac{\epsilon}{2} \tag{1}$$

$f_1$ denotes the positive part of $f$, while $f_2$ denotes the negative part. $V_E^i$ is the space between $f_i$ and $y = 0$ in $E$, i=1,2.

For i=1,2, since $V_E^i$ is measurable, there exists a Lebesgue cover of $V_E^i$ consisting finite (n+1)-dimensional cubes $J_{j,i}$, satisfying

$$m(V_E^i \triangle \bigcup_j J_{j,i}) < \frac{\epsilon}{8} \tag{2}$$

. We assume the number of $J_{j,i}s$ is $n_i$. Here and below $m(\cdot)$ denotes Lebesgue measure.

For any (n+1)-dimensional cube $J_{j,i}$, we assume

$$J_{j,i} = [a_{1,j,i}, a_{1,j,i} + b_{1,j,i}] \times [a_{2,j,i}, a_{2,j,i} + b_{2,j,i}] \times \cdots \times [a_{n+1,j,i}, a_{n+1,j,i} + b_{n+1,j,i}]$$

$$X_{j,i} = [a_{1,j,i}, a_{1,j,i} + b_{1,j,i}] \times [a_{2,j,i}, a_{2,j,i} + b_{2,j,i}] \times \cdots \times [a_{n,j,i}, a_{n,j,i} + b_{n,j,i}]$$

Note that each $J_{j,i}$ corresponds to an indicator function. we define

$$\phi_{j,i}(x) = \begin{cases} 1 & x \in X_{j,i} \\ 0 & x \notin X_{j,i} \end{cases}$$

Based on inequality (2), we have

$$\int_E |f_i - \sum_{j=1}^{n_i} b_{n+1,j,i}\phi_{j,i}|dx < \frac{\epsilon}{8} \tag{3}$$

From (1) and (3), we can prove that f can be approximated by finite weighted sum of indicator function on n-dimensional cubes. Also we have

$$\sum_{i=1}^{2} \int_E |\sum_{j=1}^{n_i} b_{n+1,j,i}\phi_{j,i}|dx = \sum_{i=1}^{2}\sum_{j=1}^{n_i} \int_E b_{n+1,j,i}\phi_{j,i}dx \tag{4}$$

$$< C + \frac{3\epsilon}{4} \tag{5}$$

Then we will show how to use ReLU network to approximate such a function.
We wish to find functions $\varphi_{j,i}$, satisfying

$$\int_{X_{j,i}} |\phi_{j,i} - \varphi_{j,i}|dx < \frac{\epsilon}{4(C + \frac{3\epsilon}{4})} \int_E |\phi_{j,i}|dx \tag{6}$$

$$= \frac{\epsilon}{4C + 3\epsilon} \int_E |\phi_{j,i}|dx \tag{7}$$

For any I $\in \{\phi_{j,i}\}$, we assume

$$I = \begin{cases} 1 & x \in X \\ 0 & x \notin X \end{cases}$$

Here,

$$X = [a_1, b_1] \times [a_2, b_2] \times \cdots \times [a_n, b_n]$$

Apparently,

$$a_j, b_j \in [-N, N], j = 1, 2, \ldots, n$$

Next we will construct a network $\mathscr{A}$ to produce a function J, satisfying

$$\int_E |I - J|dx < \frac{\epsilon}{4C + 3\epsilon} \int_E Idx \tag{8}$$

$$= \frac{\epsilon}{4C + 3\epsilon} \prod_{i=1}^{n}(b_i - a_i) \tag{9}$$

We define some notations here. We denote the network by $\mathscr{A}$, the function represented by the whole network by $F_{\mathscr{A}}$, the function represented by the $kth$ layer of the network by $F_{k,\mathscr{A}}$, the function represented by the $jth$ node in the $kth$ layer by $F_{k,j,\mathscr{A}}$, the function represented by the first $k$ layers of the network after being ReLUed by $R_{k,\mathscr{A}}$. The function represented by the $jth$ node in the $kth$ layer after ReLUed is $R_{k,j,\mathscr{A}}$. Here, without loss of generality, $R_{0,\mathscr{A}}$ denotes the input layer. The weight matrix is denoted by $A$ and the offset vector by $u$. The depth is denoted by h.

For any $\delta > 0, k = 1, 2, \ldots, n$, we can design a ReLU network $\mathscr{A}_k$ satisfying following conditions:
(1)The width of each layer of $\mathscr{A}_k$ is n+4.
(2)The depth of $\mathscr{A}$ is 3.
(3)for i=0,1,2,3, j=1,2,...,n, $R_{i,j,\mathscr{A}_k} = (x_i + N)^+$
(4)for j=n+1,n+2, all the weights related to $R_{i,j,\mathscr{A}_k}$ are 0.
(5)$R_{1,n+3,\mathscr{A}_k}$ is a function of x such that

- $0 \leq R_{1,n+3,\mathscr{A}_k}(x) \leq 1$ for any x

- $R_{1,n+3,\mathscr{A}_k}(x) = 0$ if $(x_1, \ldots, x_{k-1}) \notin [a_1, b_1] \times \cdots \times [a_{k-1}, b_{k-1}]$

- $R_{1,n+3,\mathscr{A}_k}(x) = 1$ if $(x_1, \ldots, x_{k-1}) \in [a_1 + \delta(b_1 - a_1), b_1 - \delta(b_1 - a_1)] \times \cdots \times [a_{k-1} + \delta(b_{k-1} - a_{k-1}), b_{k-1} - \delta(b_{k-1} - a_{k-1})]$

(6) $R_{3,n+3,\mathscr{A}_k}$ is a function of x such that

- $0 \le R_{4,n+3,\mathscr{A}_k}(x) \le 1$ for any x

- $R_{4,n+3,\mathscr{A}_k}(x) = 0$ if $(x_1, \ldots, x_k) \notin [a_1, b_1] \times \cdots \times [a_k, b_k]$

- $R_{4,n+3,\mathscr{A}_k}(x) = 1$ if $(x_1, \ldots, x_k) \in [a_1 + \delta(b_1 - a_1), b_1 - \delta(b_1 - a_1)] \times \cdots \times [a_k + \delta(b_k - a_k), b_k - \delta(b_k - a_k)]$

We call this shallow ReLU network Single ReLU Unit(SRU). We will explain some details of SRU. The first n+2 nodes in each layer is "memory element" of SRU while the last two is the "computation element" of SRU. The main idea of SRU is to process the function $R_{0,n+3,\mathscr{A}_k}$ to get $R_{3,n+3,\mathscr{A}_k}$.

The main idea of this process is to "chop" the function and reduce the support set of the function. See Figure 1 for a simulation sample when $n = 2$.

Figure 1: cube $I$ and hyper-trapezoid $J$ inside $I$

Denote

$$\mathscr{A} = \mathscr{A}_n \circ \mathscr{A}_{n-1} \circ \cdots \circ \mathscr{A}_1$$

We will show that, for any $\delta > 0$, $J = \mathscr{A}(x_1, x_2, \cdots, x_n)$ can produce exatly the same shape as the hyper-trapezoid inscribed in cube $I$ in Figure 1. For simplicity, define $\mathscr{B}_k = \mathscr{A}_k \circ \mathscr{A}_{k-1} \circ \cdots \circ \mathscr{A}_1$, here $k = 1, 2, \cdots, n$.
Examine $\mathscr{B}_1$. The input layer is identity function in every dimension.

$$R_{0,j,\mathscr{B}_1} = x_j$$

For simplicity, define $f^+ = ReLU(f)$. The first hidden layer retains the information of the input layer.

$$R_{1,j,\mathscr{B}_1} = \begin{cases} (x_j + N)^+ & j = 1, 2, \cdots, n \\ 0 & j = n+1, n+2 \\ 1 & j = n+3 \\ (x_1 - b_1 + \delta(b_1 - a_1))^+ & j = n+4 \end{cases}$$

The first n nodes remain unchanged thorough out the whole network $\mathscr{A}$, which are used to record the information of the input layer.The $(n+1)$ and $(n+2)$th node are reserved for the positive and negetive part of the whole target function respectively. In fact, the whole network $\mathscr{A}$ is constructed to simulate a single indicator function $I$, if the function $I$ is positive, then we will store the simulation result $J$ into the $(n+1)$th node. Otherwise, $J$ will be stored into $(n+2)$th node. By adding up

those simulation results in these two nodes, we can get a simulation of $\sum_{j=1}^{n_i}(-1)^{i+1}b_{n+1,j,i}\phi_{j,i}$, and thus simulates the target function. We list the result in second,third and fourth layer below.

$$R_{2,j,\mathscr{B}_1} = \begin{cases} (x_j + N)^+ & j = 1,2,\cdots,n \\ 0 & j = n+1, n+2 \\ (1 - \frac{(x_1 - b_1 + \delta(b_1 - a_1))^+}{\delta})^+ & j = n+3 \\ (x_1 - a_1)^+ & j = n+4 \end{cases}$$

$$R_{3,j,\mathscr{B}_1} = \begin{cases} (x_j + N)^+ & j = 1,2,\cdots,n \\ 0 & j = n+1, n+2 \\ (1 - \frac{(x_1 - b_1 + \delta(b_1 - a_1))^+}{\delta})^+ & j = n+3 \\ (1 - \frac{(x_1 - a_1)^+}{\delta})^+ & j = n+4 \end{cases}$$

$$R_{4,j,\mathscr{B}_1} = \begin{cases} (x_j + N)^+ & j = 1,2,\cdots,n \\ 0 & j = n+1, n+2 \\ L_1 = ((1 - \frac{(x_1 - b_1 + \delta(b_1 - a_1))^+}{\delta})^+ - (1 - \frac{(x_1 - a_1)^+}{\delta})^+)^+ & j = n+3 \\ 0 & j = n+4 \end{cases}$$

For simplicity, denote $L_k = R_{4,j,\mathscr{B}_k}$.The network $\mathscr{A}_k$ $(k = 2,\cdots,n)$ is similar to the case of $k = 1$.The input layer is the final layer in $\mathscr{B}_{k-1}$.

$$R_{1,j,\mathscr{B}_k} = \begin{cases} (x_j + N)^+ & j = 1,2,\cdots,n \\ 0 & j = n+1, n+2 \\ L_{k-1} & j = n+3 \\ (x_k - b_k + \delta(b_k - a_k))^+ & j = n+4 \end{cases}$$

$$R_{2,j,\mathscr{B}_k} = \begin{cases} (x_j + N)^+ & j = 1,2,\cdots,n \\ 0 & j = n+1, n+2 \\ (1 - \frac{(x_k - b_k + \delta(b_k - a_k))^+}{\delta})^+ & j = n+3 \\ (x_k - a_k)^+ & j = n+4 \end{cases}$$

$$R_{3,j,\mathscr{B}_k} = \begin{cases} (x_j + N)^+ & j = 1,2,\cdots,n \\ 0 & j = n+1, n+2 \\ (1 - \frac{(x_k - b_k + \delta(b_k - a_k))^+}{\delta})^+ & j = n+3 \\ (1 - \frac{(x_k - a_k)^+}{\delta})^+ & j = n+4 \end{cases}$$

$$R_{4,j,\mathscr{B}_k} = \begin{cases} (x_j + N)^+ & j = 1,2,\cdots,n \\ 0 & j = n+1, n+2 \\ L_k = (\frac{(x_k - b_k + \delta(b_k - a_k))^+}{\delta} - (1 - \frac{(x_k - a_k)^+}{\delta})^+ & j = n+3 \\ 0 & j = n+4 \end{cases}$$

For each k, we "chop" two sides in the kth dimension. Finally, we get the shape J in Figure 3.It is stored in the (n+3)th node as $L_n$ in the last layer of $\mathscr{A}$. We then use a single layer to record it in the (n+1)th or the (n+2)th node, and reset the last two nodes to zero. Now the network is ready to simulate another (n+1)-dimensional cube. The whole construction process is shown in Figure 4.

Using this construction, we can simulate $I$ by $J$, which is produced by network $\mathscr{A}$. Note that, as $\delta$ approaches 0, the simulation error w.r.t $L_1$ distance converges to 0.

Next we will find a value of $\delta$ to fit the need of our proof. See Figure 3. The side length of small square on the top surface is $1 - 2\delta$ as the side length of the top surface. We will select a suitable $\delta > 0$, satisfying $\int_X |I - J|dx < \frac{\epsilon}{4C+3\epsilon}\int_E |I|dx$.
Denote

$$X_0 = [a_1 + \delta(b_1 - a_1), b_1 - \delta(b_1 - a_1)] \times \cdots \times [a_n + \delta(b_n - a_n), b_n - \delta(b_n - a_n)]$$

Figure 2: The whole process to simulate a cube;every four layers are used to reshape one dimension of the cube(seperated by thick lines)

Notice that $I - J = 0$ on $X_0$, and the maximum value of $I - J$ on $X$ is 1. Thus,

$$\int_X |I - J| dx < \int_X \mathbf{1}_{x \in X \setminus X_0} dx \tag{10}$$

$$= (1 - (1 - 2\delta)^n) \prod_{i=1}^n (b_i - a_i) \tag{11}$$

Compared with (9), we set

$$\delta = \frac{1 - (1 - \frac{\epsilon}{4C+3\epsilon})^{\frac{1}{n}}}{2} \tag{12}$$

Then we have

$$\int_X |I - J| dx < \frac{\epsilon}{4C + 3\epsilon} \prod_{i=1}^n (b_i - a_i) \tag{13}$$

Satisfies

$$\int_X |I - J| dx < \frac{\epsilon}{4C + 3\epsilon} \int_E |I| dx$$

Thus, for $i = 1, 2; j = 1, 2, \cdots, n_i$, $\phi_{j,i}$ can be approximated by network function $\mu_{j,i}$. Satisfies

$$\int_E |\phi_{j,i} - \varphi_{j,i}| dx < \frac{\epsilon}{4C + 3\epsilon} \int_E \phi_{j,i} dx$$

Sum those equations up, combined with (7), we have

$$\sum_{i=1}^{2}\sum_{j=1}^{n_i}\int_E |(-1)^{i+1}b_{n+1,j,i}(\phi_{j,i}-\mu_{j,i})|dx < \frac{\epsilon}{4C+3\epsilon}\sum_{i=1}^{2}\sum_{j=1}^{n_i}\int_E b_{n+1,j,i}\phi_{j,i}dx \qquad (14)$$

$$\leq \frac{\epsilon}{4C+3\epsilon}*(C+\frac{3\epsilon}{4}) \qquad (15)$$

$$= \frac{\epsilon}{4} \qquad (16)$$

Thus, we have the approximation of cubes $J_{j,i}$. Next we show how to combine those approximation functions together by network. There are $n_1$ positive cubes, corresponding to $n_1$ positive functions $\mu_{i,1}$;$n_2$ negative cubes, correspond to $n_2$ negative functions $\mu_{j,2}$. The detailed network is shown in Figure 3.

Figure 3: The final process to simulate target function;every shown layer is the (n+1) and (n+2)th node in the last layer in Figure 4, which represent the simulation of a single cube. This figure shows the process of adding those functions up to get the function we want. Notice that except for the output layer, every result is nonnegative in the process and is produced by RELU activator. For simplicity, we just omit the RELU mark in the graph.

Finally, we have $g \triangleq \sum_{i=1}^{2}\sum_{j=1}^{n_i}(-1)^{i+1}b_{n+1,j,i}\mu_{j,i}dx$. $f_0$ is the result function produced by our designed network. Combined with (1),(3),(11), we have

$$\int_{R^n}|f-g|dx \tag{17}$$

$$< \int_{R^n}|f-(f_1-f_2)|dx + \sum_{i=1}^{2}\int_{E}|f_i - \sum_{j=1}^{n_i}(-1)^{i+1}b_{n+1,j,i}\phi_{j,i}|dx$$

$$+ \sum_{i=1}^{2}\sum_{j=1}^{n_i}\int_{E}|(-1)^{i+1}b_{n+1,j,i}(\phi_{j,i}-\mu_{j,i})|dx \tag{18}$$

$$< \frac{\epsilon}{2} + 2*\frac{\epsilon}{8} + \frac{\epsilon}{4} \tag{19}$$

$$= \epsilon \tag{20}$$

Thus, $g$ is the function we need in the theorem.

$\square$

## B    Proof of Theorem 2

The proof is long and complicated, so we firstly define some notations for convenience afterwards. We denote the network by $\mathscr{A}$, the function represented by the whole network by $F_{\mathscr{A}}$, the function represented by the $kth$ layer of the network by $F_{k,\mathscr{A}}$, the function represented by the $jth$ node in the $kth$ layer by $F_{k,j,\mathscr{A}}$, the function represented by the first $k$ layers of the network after being ReLUed by $R_{k,\mathscr{A}}$. Here, without loss of generality, $R_{0,\mathscr{A}}$ denotes the input layer. We define

Condition 1: $d_m = n$ and the widths of all the layers except the output layer are n.

Obviously other cases where $d_m \leq n$ are just special cases of this setting. The weight matrix of each layer is denoted by $A_d$ and the offset vector by $u_d$ where d is the number of layer. The depth is denoted by h.
Here we will introduce 2 definitions inspired by *Benefits of depth in neural networks* (Telgarsky ,2016).

**Definition 1**:    A set $X \subset R^n$ is a *linear block* if there exist t linear functions $(q_i)_{i=1}^{t}$, and m tuples $(U_j, L_j)_{j=1}^{m}$ where $U_j$ and $L_j$ are subsets of [t](where [t]:=1,...,t), such that $\vec{x} \in X$ is equivalent to

$$(\Pi_{i \in L_j}1[q_i(v) < 0])(\Pi_{i \in U_j}1[q_i(v) \geq 0]) = 1$$

**Definition 2**:    A function f:$R^k \to R$ is $(t,\alpha,\beta) - sa((t,\alpha,\beta) - semi - algebraic)$ if there exist t polynomials $(q_i)_{i=1}^{t}$ of degree $\leq \alpha$, and m triples $(U_j, L_j, p_j)_{j=1}^{m}$ where $U_j$ and $L_j$ are subsets of [t](where [t]:=1,...,t) and $p_j$ is a polynomial of degree $\leq \beta$, such that
$$f(v) = \Sigma_{j=1}^{m}p_j(v)(\Pi_{i \in L_j}1[q_i(v) < 0])(\Pi_{i \in U_j}1[q_i(v) \geq 0])$$

We can see Theorem 2 is a direct conclusion of Lemma 1 as follows:

**Lemma 1**:    Consider a function $F_{\mathscr{A}}$ represented by a relu neural network $\mathscr{A}$ where $d_m \leq n$, the following equation holds.
$$\int_{R^n}|F_{\mathscr{A}}(\vec{x})|d\vec{x} = 0 \ or + \infty$$

We define assumption 1 here.

Assumption 1:

$$\int_{R^n}|F_{\mathscr{A}}(\vec{x})|d\vec{x} < +\infty$$

We will prove that if assumption 1 holds,

$$\int_{R^n}|F_{\mathscr{A}}(\vec{x})|d\vec{x} = 0$$

, which is equivalent to Lemma 1. To prove Lemma 1, we need Lemma 2.

**Lemma 2**: For any given $\mathscr{A}$ where assumption 1 and Condition 1 hold and any $k \in \{0, 1, 2, \ldots, h-1\}$, there exists a *linear block* $X_k$ which satisfies following conditions:

$S_1(k)$: $X_k$ is convex.

$S_2(k)$: For any $\vec{x} \notin X_k$, $F_{\mathscr{A}}(\vec{x}) = 0$

$S_3(k)$: For any $\vec{x}$ in $B(X_k)$, $F_{\mathscr{A}}(\vec{x}) = 0$, where $B(X_k)$, the boundary set of $X_k$, is defined as $\{\vec{x} : for\ any\ \epsilon > 0, \exists \vec{u} \in X_k, \vec{v} \notin X_k s.t. ||\vec{u} - \vec{x}|| < \epsilon, ||\vec{v} - \vec{x}|| < \epsilon, \}$

$S_4(k)$: There exists a matrix $H$ and a vector $\vec{b}$ such that $R_{k,\mathscr{A}}(\vec{x}) = H\vec{x} + \vec{b}$ for $\vec{x} \in X_k$

If Lemma 2 holds and assumption 1 holds, let $k = h - 1$, $F_{\mathscr{A}}$ is a linear function on its support set, a *linear block*. It is not hard to prove Lemma 1 after that. However, the proof of Lemma 2 is difficult. Before getting into the detail, we'd like to make some remark. Our conclusion may seem strange at first since $F_{\mathscr{A}}$ is like a linear function. Note we derive all these conclusions under assumption 1. Our proof actually shows that assumption 1 does not hold in most cases and the expressive power of thin neural networks is weak.

Before proving Lemma 2, we need Lemma 3 as a preparation.

Apparently, for any Relu neural network $\mathscr{A}$, there exists an $M_0$ s.t. $F_{\mathscr{A}}$ is a $(M_0,1,1)$-sa function. This means that there exists an M s.t. $R^n$ can be partitioned into M *linear blocks* where $F_{\mathscr{A}}$ is a linear function in each block. Furthermore, $F_{\mathscr{A}}$ must be a Lipschitz function in each block. Since $F_{\mathscr{A}}$ is continuous in $R^n$, it is a Lipschitz function in $R^n$, which means there exists an L s.t.

$$|F_{\mathscr{A}}(\vec{x}) - F| \leq L||\vec{x} - \vec{y}||$$

for any $\vec{x}, \vec{y} \in R^n$. Then we can prove Lemma 3.

**Lemma 3**: If assumption 1 and Condition 1 hold, then for any ray X, if $F_{\mathscr{A}}(\vec{x})$ is constant in X, then

$$F_{\mathscr{A}}(\vec{x}) = 0$$

for any $\vec{x}$ in X.

Proof of Lemma 3: We assume $F_{\mathscr{A}}$ is L-Lipschitz. For simplicity, let $v = F_{\mathscr{A}}(X)$ and assume $v \geq 0$ without loss of generality. Then we define a set $X^+ = \{\vec{a} : \exists \vec{x} \in X s.t. ||\vec{x} - \vec{a}|| \leq \frac{v}{2L}\}$. Apparently, $F_{\mathscr{A}}(\vec{x}) \geq v/2$ for any $\vec{x} \in X^+$ and the volume of $X^+$ is $+\infty$. Thus,

$$\int_{R^n} |F_{\mathscr{A}}(\vec{x})| \geq \int_{X^+} |F_{\mathscr{A}}(\vec{x})| \tag{21}$$

$$\geq \frac{v}{2} \int_{X^+} 1 \tag{22}$$

$$= +\infty \tag{23}$$

Then we can prove Lemma 2.

Proof of Lemma 2: We prove this lemma with mathematical induction.

**Basis:** The k=0 case is simple. We let $X_0 = R^n$. It is easy to verify that $S_i(0)$ holds for i=1,2,3,4.

**Inductive step:** Given that $S_i(k)$ holds for i=1,2,3,4, we will prove that $S_i(k+1)$ holds for i=1,2,3,4 too. Let $X_{k+1} = \{\vec{x} : \vec{x} \in X_k \ and\ for\ any\ j = 1, 2, \ldots, n,\ F_{k+1,j,\mathscr{A}}(\vec{x}) > 0\}$. Apparently, $X_{k+1}$ is a *linear block* which is a subset of $X_k$. We will prove $X_{k+1}$ satisfies $S_i(k+1)$ for i=1,2,3,4.

Based on $S_4(k)$, it is easy to see $F_{k+1,\mathscr{A}}$ is a linear function on $X_k$. There exist a $n \times n$ matrix $W_{k+1}$ and a $n \times 1$ vector $b_{k+1}$ such that on $X_k$

$$F_{k+1,j,\mathscr{A}}(\vec{x}) = W_{k+1}\vec{x} + \vec{b_{k+1}}$$

. We define

$$P_{k+1,i} = \{\vec{x} : W_{k+1}(i,)\vec{x} + \vec{b_{k+1}}(i) > 0\}$$

for $i \in [n]$. Thus
$$X_{k+1} = \cap_{i=1}^{n} P_{k+1,i} \cap X_k$$
Note $P_{k+1,i}$ is convex and $X_k$ is convex based on $S_1(k)$. Thus $X_{k+1}$ is convex and so that $S_1(k+1)$ holds.

Now we are going to prove $S_2(k+1)$ holds. For any $\vec{x} \in X_k \backslash X_{k+1}$, there exists $j(\vec{x}) \in [n]$, such that $F_{k+1,j(\vec{x}),\mathscr{A}}(\vec{x}) \leq 0$. Note $j(\vec{x})$ depends on $\vec{x}$, but we write it as $j$ for simplicity.

Since $W_{k+1}$ is an $n \times n$ matrix, there must exist an n-dimensional vector $\vec{\alpha}(\vec{x}) \neq 0$ such that $\vec{\alpha}(\vec{x}) \perp W_{k+1}(i,) \ i \in [n], i \neq j$. Note, $\vec{\alpha}(\vec{x})$ depends on $\vec{x}$, however, we write it as $\vec{\alpha}$ for simplicity. We assume $W_{k+1}(j,)\vec{\alpha} \leq 0$. If it does not hold, we substitute $-\vec{\alpha}$ for $\vec{\alpha}$. Then we consider the following set
$$IRX_{\vec{x}} = \{\vec{c} : \vec{c} = \vec{x} + t\vec{\alpha} \in X_k, t \geq 0\}$$
, the intersection of $X_k$ and the ray corresponding to $\vec{\alpha}$ and $\vec{x}$. By $S_1(k)$, $X_k$ is convex. Obviously, the ray corresponding to $\vec{\alpha}$ and $\vec{x}$ is also convex. Thus $IRX_{\vec{x}}$ is a convex set and so that a continuous part of a ray. For any $\vec{y} \in IRX_{\vec{x}}$ and any $i \in [n], i \neq j$,

$$F_{k+1,i,\mathscr{A}}(\vec{y}) = W_{k+1}(i,)(\vec{x} + t\vec{\alpha}) + \vec{b_{k+1}}(i) \tag{24}$$
$$= W_{k+1}(i,)\vec{x} + \vec{b_{k+1}}(i) \tag{25}$$
$$= F_{k+1,i,\mathscr{A}}(\vec{x}), \tag{26}$$

Thus, for $i \in [n], i \neq j$

$$R_{k+1,i,\mathscr{A}}(\vec{y}) = Relu(F_{k+1,i,\mathscr{A}}(\vec{y})) \tag{27}$$
$$= Relu(F_{k+1,i,\mathscr{A}}(\vec{x}))) \tag{28}$$
$$= R_{k+1,i,\mathscr{A}}(\vec{x}) \tag{29}$$

Besides, for any $\vec{y} \in IRX_{\vec{x}}$, when i=j,

$$F_{k+1,i,\mathscr{A}}(\vec{y}) = W_{k+1}(i,)(\vec{x} + t\vec{\alpha}) + \vec{b_{k+1}}(i) \tag{30}$$
$$\leq W_{k+1}(i,)\vec{x} + \vec{b_{k+1}}(i) \tag{31}$$
$$= F_{k+1,i,\mathscr{A}}(\vec{x}) \tag{32}$$
$$\leq 0 \tag{33}$$

Thus,when i=j,

$$R_{k+1,i,\mathscr{A}}(\vec{y}) = Relu(F_{k+1,i,\mathscr{A}}(\vec{y})) \tag{34}$$
$$= 0 \tag{35}$$
$$= Relu(F_{k+1,i,\mathscr{A}}(\vec{x}))) \tag{36}$$
$$= R_{k+1,i,\mathscr{A}}(\vec{x}) \tag{37}$$

In general, we find $R_{k+1,\mathscr{A}}$ is constant on $IRX_{\vec{x}}$. Therefore $F_{\mathscr{A}}$ is constant on $IRX_{\vec{x}}$. We define
$$T = sup\{t : \vec{x} + t\vec{\alpha} \in IRX_{\vec{x}}\}$$

Since $IRX_{\vec{x}}$ is a continuous part of a ray, $\{t : \vec{x} + t\vec{\alpha} \in IRX_{\vec{x}}\}$ is an interval.

If $T = +\infty$, then $IRX_{\vec{x}}$ is a ray and thus we can conclude $F_{\mathscr{A}}(\vec{x}) = 0$ by using Lemma 3.

If $T < +\infty$, for any $\epsilon > 0$, there exist $T_1, T_2$ such that

$$T - \epsilon < T_1 < T < T_2 < T + \epsilon \tag{38}$$
$$\vec{x} + T_1\vec{\alpha} \in X_k \tag{39}$$
$$\vec{x} + T_2\vec{\alpha} \notin X_k \tag{40}$$

By the definition of $B(X_k)$, $\vec{x} + T\vec{\alpha} \in B(X_k)$. By $S_3(k)$,
$$F_{\mathscr{A}}(\vec{x} + T\vec{\alpha}) = 0$$
. On the other hand, $F_{\mathscr{A}}$ is constant on $IRX_{\vec{x}}$. Because of continuity it is constant on
$$\overline{IRX_{\vec{x}}} = IRX_{\vec{x}} \cup \{\vec{y} : for \ any \ \epsilon > 0, \exists \vec{u} \in IRX_{\vec{x}}, ||\vec{y} - \vec{u}|| < \epsilon\}$$

Obviously, $\vec{x} + T\vec{\alpha} \in \overline{IRX_{\vec{x}}}$. Thus,

$$F_{\mathscr{A}}(\vec{x}) = F_{\mathscr{A}}(\vec{x} + T\vec{\alpha})$$

Since $F_{\mathscr{A}}(\vec{x} + T\vec{\alpha}) = 0$,then

$$F_{\mathscr{A}}(\vec{x}) = 0$$

In all, for any $\vec{x} \in X_k \backslash X_{k+1}$, if assumption 1 holds, $F_{\mathscr{A}}(\vec{x}) = 0$. Besides, since for any $\vec{x} \in X_k^c$, $F_{\mathscr{A}}(\vec{x}) = 0$, then $S_2(k+1)$ holds.

Because $F_{\mathscr{A}}$ is continuous and $S_2(k+1)$ holds, we can easily find $S_3(k+1)$ holds.

By the definition of $X_{k+1}$,

$$F_{k+1,i,\mathscr{A}}(\vec{x}) > 0, for\ any\ i \in [n]\ and\ \vec{x} \in X_{k+1}$$

. Thus,on $X_{k+1}$,

$$R_{k+1,i,\mathscr{A}}(\vec{x}) = Relu(F_{k+1,i,\mathscr{A}}(\vec{x})) \tag{41}$$
$$= F_{k+1,i,\mathscr{A}}(\vec{x}) \tag{42}$$
$$= W_{k+1}\vec{x} + \vec{b_{k+1}} \tag{43}$$

It is a linear function. $S_4(k+1)$ holds.

We finish the proof of Lemma 2.

Proof of Lemma 1: If assumption 1 holds, by setting $k = h - 1$ in Lemma 3, we find there exists a *linear block* $LBX = X_k$ such that

- LBX is convex.
- $F_{\mathscr{A}}(\vec{x}) = 0$ *for any* $\vec{x} \notin LBX$ *or* $\vec{x} \in B(LBX)$
- $R_{h-1,\mathscr{A}}$ is a linear function on LBX.

Since

$$F_{\mathscr{A}} = A_h R_{h-1,\mathscr{A}} + u_h$$

, $F_{\mathscr{A}}$ is a linear function on LBX. As $F_{\mathscr{A}} = 0$ outside LBX, to finish the proof we just need to prove that for any $\vec{x} \in LBX$, $F_{\mathscr{A}}(\vec{x}) = 0$. For any $\vec{x} \in LBX$, let

$$L_{\vec{x}} = \{a\vec{x}, a \in R\}$$

and

$$IL_{\vec{x}} = L_{\vec{x}} \cap LBX$$

Since LBX and $L_{\vec{x}}$ are both convex, $IL_{\vec{x}}$ is convex. Thus there exists an interval A such that

$$t\vec{x} \in IL_{\vec{x}} \Leftrightarrow t \in A$$

Apparently, $F_{\mathscr{A}}(t\vec{x})$ is a linear function of t on A. Define

$$a = inf\ A$$
$$b = sup\ A$$

.

If $a > -\infty, b < +\infty$,then $a\vec{x}, b\vec{x} \in B(LBX)$. Thus

$$F_{\mathscr{A}}(a\vec{x}) = F_{\mathscr{A}}(b\vec{x}) = 0$$

. Since $F_{\mathscr{A}}(t\vec{x})$ is a linear function,

$$F_{\mathscr{A}}(\vec{x}) = 0$$

If $a > -\infty, b = +\infty$ or $a = -\infty, b < +\infty$, we assume $a = -\infty, b < +\infty$ without loss of generality. Then $F_{\mathscr{A}}(b\vec{x}) = 0$. If $F_{\mathscr{A}}(\vec{x}) \neq 0$, because of the linearity of $F_{\mathscr{A}}$

$$lim_{t \to -\infty} F_{\mathscr{A}}(t\vec{x}) = +\infty\ or\ -\infty$$

Since $F_{\mathscr{A}}(\vec{x})$ is Lipschitz, it contradicts with $\int_{R^n} |F_{\mathscr{A}}(\vec{x})| < +\infty$. So $F(\vec{x}) = 0$

If $a = -\infty, b = +\infty$, we can prove $F_{\mathscr{A}}(\vec{x}) = 0$ in a similar way.

In general, $F_{\mathscr{A}}(\vec{x}) = 0$ for any $\vec{x} \in R^n$ if assumption 1 holds.

Then obviously Theorem 2 is a direct result of Lemma 1.

## C Proof of Theorem 3

*Proof.* We denote the input by $\vec{x} = (x_1, x_2, ..., x_n)$, and the value of the first layer's nodes of $A$ by $y = (y_1, y_2, ..., y_m)$, here $m < n$ and let

$$y_i = (b_i + \sum_{j=1}^{m} a_{ij}x_j)^+$$

where $i = 1, 2, \cdots, n$, $j = 1, 2, \cdots, m.b_i$ and $a_{ij}$ are parameters of $A$. Since $m < n$, there exists a non-zero vector $x_0$ in $R_0^n$, which satisfies

$$\vec{x}_0 \perp span\{b_1 + \sum_{j=1}^{j=m} a_{1j}x_j, \cdots, b_n + \sum_{j=1}^{j=m} a_{nj}x_j\}$$

Since changes along $x_0$ don't affect the first layer of network $A$: $F_A$, which is determined by the first layer of $A$ itself, it is constant along $\vec{x}_0$ as a result. Thus $F_A$ must be constant along some fixed direction $x_0$.

Now we can prove that: given f and a fixed unit vector $x_0$, we have a positive $\epsilon$ that for all continuous $F$ which is constant along the direction $x_0$, the $L^1$ distance between $f$ and $F$ is lower bounded by $\epsilon$. Pick two points $a_0$ and $b_0$ along $x_0$ that $f(a_0) < f(b_0)$, due to the continuity of $f$, there exists positive $r$ and $c$ that for all $a$ in $U(a_0, r)$ and $b$ in $U(b_0, r)$, $f(b) - f(a) > c$. Let the lebesgue-measure of $U(a_0, r)$ be $V$, with the triangle inequality $|f(b) - F(b)| + |f(b - b_0 + a_0) - F(b - b_0 + a_0)| > f(b) - f(b - b_0 + a_0) > c$, we can see there exists such an $\epsilon$ which is $>= Vc$.

Then treat $\epsilon$ as a function of $x_0$. Since $\epsilon$ is positive and continuous because $f$ and $F$ are continuous and have compact domain (so any such $F$ is uniformly continuous, then 'rotating' $F$ by a small angle guarantees a small uniform difference, one can easily see $\epsilon$ is continuous now), it has a lower bound over all unit vector $x_0$. Denote this lower bound as $\epsilon^*$, $\epsilon^*$ must be positive because the set of all unit vector $x_0$ is a compact set (see it as the surface of unit ball). Since $F_A$ must be constant along some direction, $\epsilon^*$ is the desired universal constant for all $F_A$.

$\square$

## D Proof of Theorem 4

We first prove the case with input dimension $n = 1$, then the extension to $n > 1$ cases is trivial.

*Proof.* We will choose $2k^4$ different points $x^{(1)}, x^{(2)}, \ldots, x^{(2k^4)} \in R$ and consider functions represented by ReLU network on them. Here,

$$x^{(i+2k^2j)} = 2j + 1 - \frac{2k^2 - i}{4k^2}, i = 1, 2, \ldots, 2k^2, j = 0, 1, \ldots, k^2 - 1$$

For any ReLU network $\mathscr{A}$, we define a $2k^4$-dimensional vector

$$f_{\mathscr{A}} = (F_{\mathscr{A}}(x^{(1)}), F_{\mathscr{A}}(x^{(2)}), \ldots, F_{\mathscr{A}}(x^{2k^4}))$$

We will begin our proof by introducing 2 lemmas.

**Lemma 4:** We define

$$E_0 = \{(a^{(1)}, ..., a^{(2k^4)}) : 0 < a^{(i+2k^2j)} < \frac{1}{2}a^{(i+1+2k^2j)}, i = 1, 2, ..., 2k^2 - 1, j = 0, 1, ..., k^2 - 1\}$$

$$E_w = \{f_{\mathscr{A}} : \mathscr{A} \ is \ a \ ReLU \ network \ with \ width \ 2k^2, depth \ 3, input \ width \ and \ output \ width \ 1\}$$

Then

$$E_0 \subset E_w$$

proof of Lemma 4:

For any $f \in E_0$, we will fabric a ReLU network $\mathscr{A}$ with width $2k^2$ and depth 3 such that $f = f_{\mathscr{A}}$. Firstly, it is easy to choose appropriate first layer weights and bias to make

$$R_{1,\mathscr{A}} = ((x)^+, (x-1)^+, \ldots, (x - 2k^2 + 1)^+)'$$

Denote the weights and bias of kth layer by $W_{k,\mathscr{A}}$ and $B_{k,\mathscr{A}}$. $W_{k,\mathscr{A}}$ is a matrix and $B_{k,\mathscr{A}}$ is a vector such that

$$F_{k+1,\mathscr{A}} = W_{k,\mathscr{A}} R_{k,\mathscr{A}} + B_{k,\mathscr{A}}$$

Define $F_{2,i,\mathscr{A}}$ to be the function at the $ith$ node in the second layer, which is a piecewise linear function which is linear between any integral points on the x-axis. It satisfies:

$$F_{2,i,\mathscr{A}}(x^{(i+2k^2j)}) = a^{(i+2k^2j)} \quad i = 1; j = 0, 1, ..., k^2 - 1$$

$$F_{2,i,\mathscr{A}}(x^{(i+2k^2j)}) = a^{(i+2k^2j)} - 2a^{(i-1+2k^2j)} \quad i = 2; j = 0, 1, ..., k^2 - 1$$

$$F_{2,i,\mathscr{A}}(x^{(i+2k^2j)}) = a^{(i+2k^2j)} - 2a^{(i-1+2k^2j)} + a^{(i-2+2k^2j)} \quad i = 3, 4, ..., 2k^2; j = 0, 1, ..., k^2 - 1$$

and that

$$F_{2,i,\mathscr{A}}(2j + 1 - \frac{2k^2 - i + 1}{4k^2}) = 0, i = 1, 2, ..., 2k^2; j = 0, 1, ..., k^2 - 1$$

Together with the linearity between integral points on the x-axis, the function represented by the $ith$ node can be uniquely decided. Then we activate those functions by RELU, and add them up to get the final output $f_{\mathscr{A}}$. One can easily check that

$$f_{\mathscr{A}} = (a^{(1)}, ..., a^{(2k^4)})$$

Combined with the definition of $E_0$ and $E_w$, we have:

$$E_0 \subset E_w$$

Define

$$\mathscr{F}_k = \{\mathscr{A} : \mathscr{A} \text{ is a ReLU network with width } 2k^2, depth 3, input and output dimension 1; f_{\mathscr{A}} \in E_0\}$$

**Lemma 5:** For any k$\geq$5, only a 0 measure set(Lebesgue measure on the weight and bias space) of the networks in $\mathscr{F}_k$ can be equaled by a deep network whose width $\leq k^{\frac{3}{2}}$ and depth $\leq k + 2$.

proof of Lemma 5:

We prove a stronger statement: only a 0 measure set(Lebesgue measure on the weight and bias space) of the networks in $\mathscr{F}_k$ can be equaled on specific $2k^4$ different points $x^{(1)}, x^{(2)}, \ldots, x^{(2k^4)}$,by a deep network whose width $\leq k^{\frac{3}{2}}$ and depth $\leq k + 2$. Notice the fact that a network with width $d$ and depth $h$ has degree of freedom $= d^2(h-2) + d(h-1) + 2d + 1$. Define $\mathscr{B}$ to be one of the deep networks, with width $d \leq k^{\frac{3}{2}}$ and depth $h \leq k + 2$. Let $g_0$ be the function mapping the parameters of the deep network to $f_{\mathscr{B}}$:

$$g_0 : R^{d^2(h-2)+d(h-1)+2d+1} \rightarrow R^{2k^4}$$

$$g_0(all\ parameters) = f_{\mathscr{B}}$$

.

When $d \leq k^{\frac{3}{2}}$ and $h \leq k + 2$, the degree of freedom of the deep network $\leq k^4 + k^3 < 2k^4$, and $g_0$ is $C_1$-derivable almost everywhere. Thus, $B$: the set of all $\beta$, which is the solution space of $g_0$ has a zero measure in $R^{2k^4}$ according to Differential Homeomorphism Theorem. In fact, we can implement the original mapping to a new function $g_1$

$$g_1 : R^{2k^4} \rightarrow R^{2k^4}, g_1(all\ parameters, p_1, ...) = g_0(all\ parameters)$$

in the way of adding variables $p_1, p_2, ..., p_{2k^4 - d^2(h-2)-d(h-1)-2d-1}$ which have no effect on the value of $F$, then the Jacobian of $g_1$ is zero now because the differential of $F$ to $p_i$s is 0, thus by the transform formulation of integration, the measure of the range is zero.

$$m(range(g_1)) = \int_{R^{2k^4}} dg_1 = \int_{R^{2k^4}} \frac{\partial g_1}{\partial \vec{x}} d\vec{x} = 0$$

It's obvious that $m(E_0) > 0$, so $E_0 \cap range(g_1)$ is a negligible subset in $E_0$ and as a result only a negligible set of the functions in this family of wide networks can be equaled by such deep networks.

Then because all parameters in these deep networks are bounded, we can extend the difference on finite points to integration on input domain.

Apparently, the shape of such a deep network can be denoted by a vector whose $m^{th}$ entry denotes the width of the $m^{th}$ layer except for the output layer. We denote the shape vector of a network $N$ by S(N). Thus for all networks with $h \leq k + 2$ and $d_m \leq k^{1.5}$,

$$S(N) \in V$$

$$here \quad V = \{(w_1, w_2, ..., w_h)| \quad h \leq k + 2 \quad and \quad w_m \leq k^{1.5} \quad for \quad any \quad m\}$$

Denote the all elements of $V$ by $\{V_j\}$, we only need to prove Lemma 6 as followed,then $n = 1$ case is proved directly by setting $\epsilon = min_{j \leq |V|}\{\epsilon_j\}$:

**Lemma 6:** For any wide network $N_w$ which can't be equaled by deep networks with width $\leq k^{1.5}$ and depth $\leq k + 2$ as above, there exists a $\epsilon_j > 0$ for all deep network $N_d$ with $S(N) = V_j$ satisfies

$$\int_0^{2k^2} (N_d(x) - N_w(x))^2 \geq \epsilon_j$$

Set $\epsilon_j = inf\{\int_0^{2k^2} (N_d(x) - N_w(x))^2, S(N_d) = V_j\}$ We are going to prove $\epsilon_j > 0$. With the conclusion of inequability above and continuity of the function $N_d$ and $N_w$, we know for any

$$S(N_d) = V_j, \int_0^{2k^2} (N_d(x) - N_w(x))^2 > 0$$

Thus, if $\epsilon_j = 0$ There must be a sequence $N_{d_i}$ satisfies

$$\int_0^{2k^2} (N_{d_i}(x) - N_w(x))^2 < \frac{1}{i}$$

Since every bounded sequence(here the assumption of parameters' bound is used, so for different choice of $b$, $\epsilon$ changes) has a convergent subsequence and parameters of a network are bounded as well, we can find a subsequence $N_{d_{i_j}}, j = 1, 2, \ldots$,every parameter of which converges. We define the network they converge to is $\tilde{N}$. Then for any x, $(N_{d_{i_j}}(x) - N_w(x))^2$ converges to $(\tilde{N}(x) - N_w(x))^2$. Besides, the values of them are uniformly bounded. Thus, with Dominated Convergence Theorem, we can find

$$\int_0^{2k^2} (\tilde{N}(x) - N_w(x))^2$$

$$= \int_0^{2k^2} \lim_{j \to \infty} (N_{d_{i_j}}(x) - N_w(x))^2$$

$$= \lim_{j \to \infty} \int_0^{2k^2} (N_{d_{i_j}}(x) - N_w(x))^2$$

$$= 0$$

This causes contradiction to our conclusion of inequability above. So $\epsilon_i > 0$ and we are finished with the proof of the case with $n = 1$. $\qquad \square$

For cases with $n > 1$, we denote these $n$ inputs by $x_1, ..., x_n$. We construct the same wide network for $x_1$ only and ignore other inputs(set the weights from them to the first later to be 0). Our wide network still has width $2k^2$ and depth 3, and for any deep network with width $\leq k^{1.5}$ and depth $\leq k + 2$ all our results above hold as well (for the choice of the prechosen $2k^4$ points, their value on $x_2, ..., x_n$ can be arbitrary). The whole proof is finished now.