[Reviews · NeurIPS 2017]

Reviewer 1



The paper is on width as a complexity measure for neural networks. The activation function is assumed to be ReLU, and approximation is measured by the L^1 distance over R^n. Positive and negative results are given on approximability, the width/depth trade-off is studied, and some experimental results are given. A proof outline is given for the positive result only (Theorem 1). Detailed proofs are given in the Appendix. The negative results (Theorem 2 and 3) seem somewhat unusual. Theorem 2 states inapproximability of *every* nontrivial function by width n networks. One wonders whether this is mainly caused by the fact that ReLU and L^1 distance over R^n is an unnatural combination to consider? This, of course is not a well-defined question, but certainly some justification of this choice should be given in the paper. Also, some intuition should be given for the proof (which uses concepts from [17] and gives some kind of normal form). Theorem 3 is formulated in a strange way: $\epsilon$ has no role in the statement. The theorem simply says that under the stated conditions the target function and the function computed by the network (both continuous) are different on the cube. Thus the theorem only gives a sufficient condition for a continuous function to be not computable exactly by the given class of networks. In this case the proof (given in Appendix C) is simple, but the short proof contains several typos: the limits for $i$ and $j$ are exchanged, and the argument of the span is wrong (the members of the list are not vectors). This is somewhat worrisome with respect to the longer proofs which I could not inspect in detail. It seems that the interpretation of the experiments is somewhat problematic, as `exceeding the bound of Theorem 4 by a constant factor' for a few small sizes is hard to interpret. Also, the width-depth trade-off problem is about worst-case (called `existential' in the paper), so one should be careful to interpret the experimental results as evidence for the worst-case behavior. Considering the network width parameter is an interesting topic. The choice of the model (activation function and approximation criterion) is not justified in the paper, and it is not clear how much the particular results are due to this choice. The results seem interesting, and the open problem posed about the width-depth trade-off seems interesting as well.

Reviewer 2



The paper studies the expressive power of width--bounded, arbitrary--depth networks. Specifically they derive universal approximation statements for such networks. The analogue result for depth--bounded width--unbounded network received considerable attention and states that depth=2 and exponential width is enough to approximate any network. The first natural question is whether width=2 and exponential depth should be enough. Not surprisingly this is not true. But moreover the authors give a crisp threshold and show that unless width > n (n is input dimension) no universal depth theorem holds (thm 2. and 3.) I think this result is interesting, and motivates understanding the effect of width. On a positive note, they do show that for width= n+4, we can achieve universal depth theorem: Roughly the idea is to propagate the input layer, as well as compute additional neuron at each layer and sum all calculation so far: which roughly turn a large width network into a large depth network. In terms of clarity and writing: In my opinion the authors put a lot of focus on theorem 1 and not enough focus on theorem 2,3: There is no overview of these statements proofs and it is very hard to verify them. The author should give an overview of the techniques and proof. Currently these theorems are simply differed to the appendix and even there the authors simply provide some list of technical Lemmas. On a broader level, the main caveat of these universal approximation theorems is that they relate to arbitrary, very large and general classes of functions (Lebesgue integrable). The crucial property of deep networks, in terms of expressive power, is that with bounded width and poly-depth you can approximate any efficiently computable function. Nevertheless, it is still interesting to understand the approximation power of arbitrary functions on a theoretical basis. Also, the lower bound showing that with sublinear width there is no universal approximation even for exponential depth is an interesting result.

Reviewer 3



SUMMARY * This paper studies how width affects the expressive power of neural networks. It suggests an alternative line of argumentation (representation of shallow by deep narrow) to the view that depth is more effective than width in terms of expressivity. The paper offers partial theoretical results and experiments in support of this idea. CLARITY * The paper expresses the objectives clearly and is easy to follow. However, the technical quality needs attention, including confused notions of continuity and norms. RELEVANCE * The paper discusses that too narrow networks cannot be universal approximators. This is not surprising (since the local linear map computed by a ReLU layer has rank at most equal to the number of active units). Nonetheless, it highlights the fact that not only depth but also width is important. * The universal approximation results seem to add, but not that much to what was already known. Here it would help to comment on the differences. * The paper presents a polynomial lower bound on the required number of narrow layers in order to represent a shallow net, but a corresponding upper bound is not provided. * As an argument in favor of depth, the theoretical results do not seem to be conclusive. For comparison, other works focusing on depth efficiency show that exponentially larger shallow nets are needed to express deep nets. The paper also presents experiments indicating that a polynomial number of layers indeed allows deep narrow nets to express functions computable by a shallow net. OTHER COMMENTS * The paper claims that ``as pointed out in [2]: There is always a positive measure of network parameters such that deep nets can be realized by shallow ones without substantially larger size''. This needs further clarification, as that paper states `` we prove that besides a negligible set, all functions that can be implemented by a deep network of polynomial size, require exponential size in order to be realized (or even approximated) by a shallow network.'' * In line 84, what is the number of input units / how do the number of parameters of the two networks compare? * In related works, how to the bounds from [14,18] compare with the new results presented in this paper? * In line 133. Lebesgue measurable functions are not generalizations of continuous functions. * In line 134 the argumentation for L1 makes no sense. * In line 163 ``if we ignore the size of the network ... efficient for universal approximation'' makes no sense. * The comparison with previous results could be made more clear. * Theorem 2 is measuring the error over all of R^n, which does not seem reasonable for a no-representability statement. * Theorem 3 talks about the existence of an error epsilon, which does not seem reasonable for a no-representability statement. Same in Theorem 4. * In line 235 the number of points should increase with the power of the dimension. CONCLUSION The paper discusses an interesting question, but does not answer it conclusively. Some technical details need attention.